# BIM-based and AR Application Combined with Location-Based Management System for the Improvement of the Construction Performance

**Julia Ratajczak [1,2,]***, **Michael Riedl [3]** and **Dominik T. Matt [3,4]**

[1]   Faculty of Science and Technology, Free University of Bozen-Bolzano, 39100 Bolzano, Italy
[2]   Budimex SA, 01-040 Warsaw, Poland
[3]   Fraunhofer Italia Research, 39100 Bolzano, Italy; michael.riedl@fraunhofer.it (M.R.);
      dominik.matt@fraunhofer.it (D.T.M.)
[4]   Faculty of Science and Technology, Free University of Bozen-Bolzano, 39100 Bolzano, Italy
[*]   Correspondence: julia.ratajczak@natec.unibz.it

**Abstract:** The information and communication technologies (ICTs) utilization ratio in the construction industry is relatively low. This industry is characterized by low productivity, time and cost overruns in projectsdue to inefficient management processes, poor communication and low process automation. To improve construction performance, a BIM-based (BIM - (Building Information Modelling) and augmented reality (AR) application (referred to as the AR4C: Augmented Reality for Construction) is proposed, which integrates a location-based management system (LBMS). The application provides context-specific information on construction projects and tasks, as well as key performance indicators on the progress and performance of construction tasks. The construction projects are superimposed onto the real world, while a site manager is walking through the construction site. This paper describes the most important methods and technologies, which are needed to develop the AR4C application. In particular, the data exchange between BIM software and the Unity environment is discussed, as well as the integration of LBMS into BIM software and the AR4C application. Finally, the implemented and planned functionalities are argued. The AR4C application prototype was tested in a laboratory environment and produced positive feedback. Since the application addresses construction sites, a validation in semi-real scenarios with end users is recommended.

**Keywords:** BIM; augmented reality; location-based management system; digital construction; lean construction; construction management; mobile application

## 1. Introduction

McKinsey claims [1] that the construction sector creates 13% of the world's GDP (Gross Domestic Product), but its labor-productivity grew approx. 1% over two decades, compared with 2.8% growth for the total economy, and 3.6% for manufacturing. This means that over these years the construction industry could increase its value by $1.6 trillion a year, if its productivity were to catch up with the total economy. The construction industry is a project-based industry characterized by heterogeneity, extreme complexity, a fragmented supply chain and variability of trade performance. These particularities cause poor productivity, which is a big issue. Construction projects are subjected to high risk in terms of schedule deviation and cost overruns [2]. According to McKinsey [3], 98% of megaprojects have to deal with cost overruns or delays, with an average cost increase being 80% of original budget, and an average delay being 20 months behind original schedule. Katre and Ghaitidak [4] assessed factors influencing time and cost overruns of construction projects and classified a low productivity of labor as one of the critical factors. Hussin et al. [5] reported that 70% of construction projects are

affected by time overruns, with 14% of the project contract sum consumed by cost overruns, and 10% of project materials ending up as waste material. Low productivity in the construction industry and increased time and cost of projects is caused by a significant number of wastes [6]. Hussin et al. [5] outlined in his literature review that construction wastes are generated by frequent design changes, poor quality of materials, low-skilled workers, poor planning and site management practices, materials not in compliance with specifications, and other factors. In addition, some unpredictable factors, such as weather conditions, can extend the duration of certain construction tasks. According to statistics published by KPMG International in 2015 [7], only 25% of projects came within 10% of their original deadlines in the past three years in the global construction industry. This occurs because inadequate and inaccurate monitoring and control processes are practiced on worksites [8].

An effective monitoring of construction performance and progress is crucial to delivering project on time within an established budget [9]. Continuous verification of a project's status allows site managers to identify problems early and make conscious decisions in time to prevent scheduling deviations [10]. In conventional processes, the monitoring and controlling of construction processes are mostly done by paper-based or simple IT (Information Technology) tools [11]. This is a time-consuming process that requires site managers to analyze and calculate huge amount of data and fill out paper forms. In addition, site managers often do not have tools to visualize and represent the information in a simple and user-friendly manner at their disposal [12].

To provide site mangers with meaningful data on project performance and progress, it is important to introduce efficient monitoring methods. By applying the automation of field data capturing to traditional progress control, it would be possible to analyze construction status, which is fundamental to improve efficiency in construction project decision-making [9]. Site managers often are not equipped with field tools that support the automated monitoring and controlling of construction works. Beyond the monitoring of construction processes, an important aspect of performance is the quality control of performed works. According to Love at al. [13], omissions in quality controls may cause construction errors and quality degradation, which negatively affect both costs and project schedules. Therefore, insufficient management and - quality controls affect delays, project profitability and cost increase [14]. This also has relevant impacts on productivity [15].

Over the years, the construction industry has struggled with sharing information between construction project participants, which is one of the most common causes of poor performance [16]. To enhance the efficiency and productivity of construction processes, it is crucial to provide accurate and timely information on site, and to apply efficient management of the information flow, as well as improve communication. In major construction sites, information is still managed by means of paper-based documents, including construction drawings, construction logs and scheduling. This situation often leads to misunderstandings between stakeholders, construction errors and does not provide a holistic vision of the current situation, that hinders informed decision-making. A lack of information or faulty information on construction sites also increases the likelihood of errors, which can lead to a reduction of building quality [17].

To summarize, the authors identified the following main problems and deficits in the management of construction processes:

- low labor productivity, which affects time and cost overruns of construction projects;
- low productivity, caused by waste generated during construction processes such as inefficient construction planning and site management, poor quality, lack of information and ineffective control [18];
- lack of automation in monitoring and controlling of construction works, as site managers mostly use paper-based or simple IT tools, which often are not sufficient to fully control the construction progress and performance;
- lack of information, which often leads to communication issues and construction errors that translate into higher costs and schedule deviations.

Based on these assumptions, the authors initiated a research project focused on the development of a BIM-based BIM—(Building Information Modelling) and augmented reality (AR) application combined with lean construction practices (referred to as the AR4C). The project presents a solution for site managers and workers that addresses the aforementioned problems and deficits. In this research paper, the authors focus mainly on a description of the methodologies and technologies used during the development of the application, as well as its main functionalities and integration into systems. In the market, several construction management software products are available, therefore a review of these solutions is performed (Section 2). In this section, the authors highlight the main differences between their proposed solution and other commercial solutions. Section 3 describes the technological solution for field application (AR4C). To provide all of the functionalities, which are listed in Figure 1, several technologies and methodologies have been integrated into the application. Their description and implementation methods are discussed in Section 4, along with the various functionalities of the application, which are also discussed in Section 5. In Section 6, preliminary tests of the application in a laboratory environment and in real buildings are discussed. Finally, as a conclusion, the authors outline the novelty of the application, and gaps that it covers in relation to the other solutions. The advantages and limitations of the proposed solution are highlighted as well.

## 2. Review of Technological Solutions for the Construction Management

In recent years, the adoption of information and communication technologies (ICTs) in the construction industry have had a significant impact on both productivity and economic growth for construction companies [19]. However, the utilization of ICTs to automate processes is relatively low compared to other industries [20]. This is not caused by a lack of willingness of construction companies to adopt new solutions, but rather it is slowed down by several organizational and technical barriers such as a lack of skilled employees and social and habitual resistance to change, which inhibit the full digitalization and automation of construction processes.

Along with the development of technologies like building information modeling (BIM), augmented reality (AR), virtual reality (VR) and internet of things (e.g., near-field communication (NFC) and radio-frequency identification (RFID) sensors), new hardware and software tools have been introduced into the construction industry. These technologies allow the automation of construction processes, monitoring of construction works and management of information flow, as well as quality inspections. Leading commercial software companies have proposed solutions for automating construction project controls. BIM-based construction management platforms and mobile field applications enable users to plan, update and manage construction works, as well as manage documents, visualize 3D models of a project and monitor construction project's status. Some examples of this are Autodesk BIM 360 (Build, Docs and Plan) [21], Oracle Aconex Connected BIM [22], Oracle Latista [23], Dalux TwinBIM [24], Trimble Vico Office [25] and VisiLean [26]. In addition, these software products use BIM models of the project site to facilitate controlling processes, and some of them integrate AR/VR technology to visualize interactive 3D models on site. Autodesk BIM 360 cooperates with DAQRI [27] to integrate AR in order to display 3D models, information and documents using a smart helmet. Recently, Dalux released the TwinBIM application, which allows users to access to the latest project information through a 3D model and put it into a real perspective using AR.

Increasing automation of processes on site due to BIM and AR technology can improve decision-making process and provide real-time access to information. However, BIM-based and lean management software products such as Autodesk BIM 360 Plan, Trimble Vico Office and VisiLean have not yet integrated AR technology to display tasks and task related information, as well as construction progress and performances. Only the VTT Technical Research Centre has been developing an external AR application to display 3D models, which is compatible with VisiLean. On the other hand, BIM-based and AR applications such as Dalux TwinBIM and Autodesk BIM 360 Docs do not manage construction processes according to lean management practices. Instead, they just focus on providing 3D interactive models and documents on site to assist inspection and report issues. Figure 1

highlights the differences between AR4C field applications and commercial software products that are used for the construction management. It shows that the AR4C application aims to provide one field tool, which is able to support lean construction on site and the AR visualization of digital contents to streamline information delivery related to the project and the construction process.

| Functionalities | Autodesk ® BIM 360 Plan™ | Autodesk® BIM 360 Docs™ | Orade Aconex Connected BIM | Orade Latista | Dalux TwinBIM | Trimble Vico Office (not mobile app) | VisiLean | AR4C |
|---|---|---|---|---|---|---|---|---|
| • 3D model visualization | ✓ | ✓ | ✓ | ✓ | ✓ | ✓ | ✓ | ✓ |
| • 3D object filtering | ✓ | ✓ | ✓ | ✓ | ✓ | ✓ | ✓ | ✓ |
| • Visualization of 3D model in Augmented Reality (AR) | X | ✓ | X | X | ✓ | X | X¹ | ✓ |
| • 3D model superimposed on real world in AR | X | ✓ | X | X | ✓ | X | X | ✓ |
| • 3D interactive models (data displaying of each Building Information Modeling (BIM) object) | X | ✓ | ✓ | ✓ | ✓ | ✓ | ✓ | ✓ |
| • Visualization of attached documents to BIM objects | X | ✓ | ✓ | X | ✓ | X | ✓ | ✓ |
| • Notes attachment to BIM objects | ✓ | ✓ | ✓ | ✓ | ✓ | X | ✓ | ✓ |
| • Task list related to locations | ✓ | X | X | X | X | ✓ | ✓ | ✓ |
| • Instructions for the execution of construction works linked to task list | X | X | X | X | X | X | ✓ | ✓ |
| • Quality Checklists linked to tasks | X | X | X | ✓ | X | X | ✓ | ✓ |
| • Construction progress and performance tracking | ✓ | X | X | X | X | X | ✓ | ✓ |
| • Construction progress and performance reporting (dashboard) | ✓ | X | X | X | X | X | ✓ | ✓ |
| • Reporting of performance and progress Key Performance Indicators on BIM model in each location | X | X | X | X | X | X | X | ✓ |

**Figure 1.** Differences between functionalities of the AR4C application and functionalities of the other commercial software products for the construction management.

In fact, the literature review demonstrates that research projects have been carried out to provide visual information on project performance and progress. KanBIM is a BIM-based system that supports production planning and day-to-day production control on construction sites based on the last planner system. It provides visual information using building models [28]. D4 AR is an image-based modeling technique for visualizing construction progress discrepancies between as-planned and as-built using

daily progress photographs and the superimposition of reconstructed scenes over as-planned 4D models [29]. Kopsida and Brilakis [30] proposed a solution for a markerless, mobile-based AR solution that assist inspection and progress monitoring for interior finishing works by displaying a 3D as-planned BIM model and detecting differences with actual construction works. In the ACCEPT (Assistant for Quality Check during Construction Execution Processes for Energy-efficienT buildings) project, a system for construction management was developed that uses smart glasses and a smartphone to display an overlaid AR digital model and information onto a real construction environment [31]. However, in these research projects, the monitoring of project performance and progress according to lean construction methods, as well as the visualization of construction progress and performance in specific locations using AR, has not be proposed yet. Based on the review of the available technological solutions, it has been determined that an application that is able to merge BIM, AR and lean functionalities is required to maximize the improvement of construction processes on site.

## 3. Proposed Solution

This research project proposes a prototype of a BIM-based and AR application called the AR4C (Augmented Reality for Construction) that is combined with a location-based management system (LMBS) to improve performance of construction works (Figure 2). The AR4C aims to improve aspects of construction performance such as productivity, quality of construction work and information flow. Productivity will be enhanced by implementing the monitoring of construction works on a daily basis in a specific location of the project. It is planned for the AR4C application to enhance project control via the rapid identification of deviations from a project's schedule, as well as variations in performance and progress, by overlaying a 3D BIM model on the real world using AR. The quality of construction works will be increased by providing context-specific information on tasks, building components and materials anytime and anywhere on the construction site. Construction works will be verified by linking quality checklists to each construction task. Moreover, information flow will be streamlined by displaying tailored information for each construction task through 3D models and lists of construction tasks.

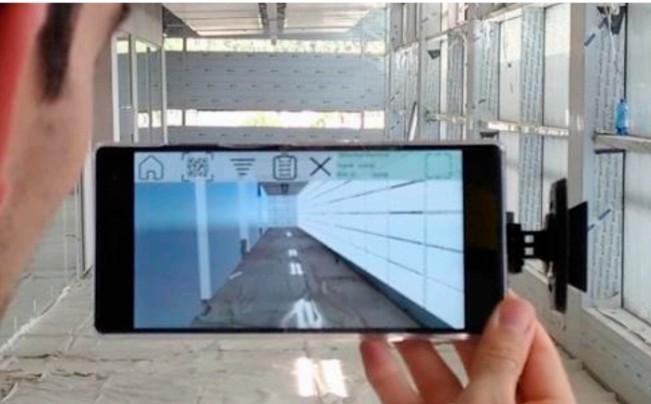

**Figure 2.** AR4C application used on the construction site.

Research activities on the AR4C project initiated in the European project ACCEPT (www.accept-project.com) are currently carried out by Fraunhofer Italia and the Free University of Bozen-Bolzano within the PhD program Sustainable Energy and Technologies.

## 4. Enabling Technologies and Methods used in the AR4C Application

### 4.1. Building Information Modeling (BIM) and Lean Constrcution

BIM is a process and technology that involves the generation and management of a digital representation of the physical and functional characteristics of a facility (e.g., a building). It is also

a shared knowledge resource for information about the facility, and its components and materials, that forms a reliable basis for decision-making processes during its life cycle [32]. According to Eastman et al. [33] (p. 1), "BIM facilitates a more integrated design and construction process that results in better quality buildings at lower cost and reduced project duration". Sacks et al. [34] and Dave [35] investigated the potential for synergy between BIM and lean construction, and tested their implementation in the field. Khanzode et al. [36] studied a conceptual framework to link virtual design and construction (VDC) with lean project delivery. From their study it was determined that lean project delivery improves VDC, if applied to appropriated stages. Sacks et al. [28] discussed the use of BIM to visualize construction processes through pull signals displayed on a 3D status board. Khemlani [37] demonstrated that the integration of lean processes combined with BIM streamlined the construction process and allowed construction companies to deliver a project on time and within its budget. Sacks et al. [28] defined and analyzed 56 interactions between lean construction and BIM, and indicated the most relevant that should be further explored. Dave [34] discussed the concept of a lean production management system that integrates BIM and provides users with construction process status. Lean construction techniques are promising for the reduction, if not the complete elimination, of non-value adding works [38]. A promising management system that applies lean construction principles is the location-based management system (LBMS), which focuses on production control based on pull controlling [19]. In LBMS, construction activities and their controls are always in reference to locations. Organization of activities by location provides information that is more comprehensive, avoids interruption between different trades and enhances constancy of the workflow [19]. It can also increase productivity and prevent production problems, which cause cascading delays and impact project durations by 10% [39]. Seppänen et al. [40] evaluated how LBMS is able to increase production rates on average by 37%, and prevent production problems by 50%.

The AR4C application integrates both BIM and LBSM. BIM is used to provide a 3D interactive model with geometrical and technical data related to components and materials as well as information on scheduled tasks in specific locations at the construction site. In regard to this, the challenge in this research project was to integrate LBMS into the BIM software (Autodesk Revit). The proper integration of LBMS enabled the visualization of construction tasks for a location defined by the site manager. In the AR4C, task scheduling is defined according to location hierarchy using respective location breakdown structure (LBS) codes. Codes can be defined based on a three-level hierarchy (Figure 3). Each location hierarchy has a different scope. The highest level (level 1) refers to locations where the structure can be built independently (e.g., individual buildings or parts of large buildings). The middle level (level 2) defines the production plans for the flow of structures, which always refers to floors. The lowest level (level 3) is used to effectively plan construction tasks at a detailed level. It is important to define locations in a way that permits the accurate monitoring of a task's progress.

The location is defined according to type of work that has to be performed (e.g., façade installation should consider orientation, while finishing works should be defined according to the type of space such as a room or apartment). In the AR4C, LBS codes are composed by combining an abbreviation of the location nomenclature at each level (e.g., BLDG1.F1.U1: Building 1–Floor 1–Unit 1). Construction tasks and their controls always refer to those locations. Moreover, a task hierarchy has been introduced in LMBS. Each of the construction tasks is defined by a work breakdown structure (WBS) code, as well as the aforementioned LBS code. The combination of both codes provides a unique nomenclature for each task, the so-called WBS/LBS code, which is used in the AR4C to identify a specific task in a specific location. WBS and LBS codes are inserted in objects of the 3D BIM model in proprieties and parameters. Firstly, a Revit shared-parameter file (Figure 4) has to be uploaded to Autodesk Revit. Afterwards, parameters of the WBS and LBS codes can be found in the 'proprieties' of each element and material, as well as the rooms of the project. When an object of the 3D model is selected, those codes are visible, and it is possible to provide the number of a WBS code and an abbreviation of an LBS code. In cases of an element composed of multiple layers (e.g., a wall), a WBS code is provided for each construction task that is related to the component. For instance, a wall component is composed

of a concrete structure, a thermal insulation layer and a brick layer. This means that three separate tasks will be scheduled in this location with different WBS codes, as shown in Figure 5. Since task names are not inserted in objects in Autodesk Revit, it is necessary to include WBS and LBS codes in a master schedule in order to link AR4C elements of the 3D model to respective tasks in a specific location. The master schedule is location-based, prepared in Microsoft Project [41] and exported as an .xml file. This file is imported into Autodesk Revit through an ACCEPT XML plugin developed by CYPE Ingenieros within the ACCEPT project. The plugin exports Revit model data with LBS and WBS codes, while at the same time adding the previously imported .xml file from the Microsoft Project. The file generated by this plugin is imported into Unity, which is the platform for creating the AR4C application. In Unity, this file enables application to: (a) display a list of task and related information, (b) display construction performance and progress Key Performance Indicators (KPIs) in a specific location for selected tasks or a group of tasks and (c) highlight task progress status on objects of the 3D model, which contain the same WBS and LBS codes as in selected tasks. In other words, information contained in this file will allow a graphical representation of where construction works should be executed and how they are progressing. While a worker is walking though the construction site with the AR4C application, they should see tasks assigned to them on the 3D model superimposed on the surroundings via AR. When they click on a task, related components/materials are highlighted on the model, and information on installation process is displayed.

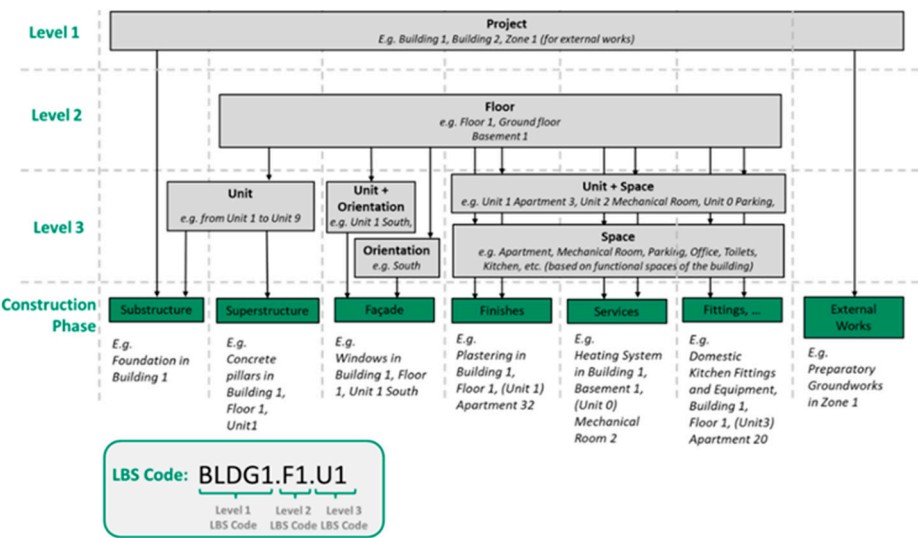

**Figure 3.** Location breakdown structure of the construction project's location hierarchy.

```
LBS-WBS Codes - Notepad
File  Edit  Format  View  Help
# This is a Revit shared parameter file.
# Do not edit manually.
*META    VERSION MINVERSION
META     2       1
*GROUP   ID      NAME
GROUP    1       Codes
*PARAM   GUID         NAME        DATATYPE        DATACATEGORY    GROUP    VISIBLE DESCRIPTION    USERMODIFIABLE
PARAM    cc18b263-5c70-4912-80fd-87e57541bd96    8_WBS Code      TEXT        1       1                       1
PARAM    d07fc366-9168-4754-995d-101e16affd6c    7_WBS Code      TEXT        1       1                       1
PARAM    22bce76d-774c-4b4c-a5c8-88fec456bcdd    5_WBS Code      TEXT        1       1                       1
PARAM    a89abb6e-7b6b-4d3e-99db-947fc4415f0e    2_WBS Code      TEXT        1       1                       1
PARAM    fb3cd67d-a788-4331-99ca-cf88820e47b7    1_WBS Code      TEXT        1       1                       1
PARAM    4d091385-3ad4-4eef-8cd9-c1c6efc82331    6_WBS Code      TEXT        1       1                       1
PARAM    8080999c-fbb4-481c-af8b-daf9bea82d36    LBS Code        TEXT        1       1                       1
PARAM    4d605eb5-ded5-4e26-a6ee-c23d2f5e79ae    4_WBS Code      TEXT        1       1                       1
PARAM    b84467dc-ccf1-492c-a9e5-02317f183c14    3_WBS Code      TEXT        1       1                       1
```

**Figure 4.** Structure of a Revit shared parameter file (.txt) with an Location Breakdown Structure (LBS) code and several Work Breakdown Structure (WBS) codes that will be displayed in Autodesk Revit once an object, material or room is selected.

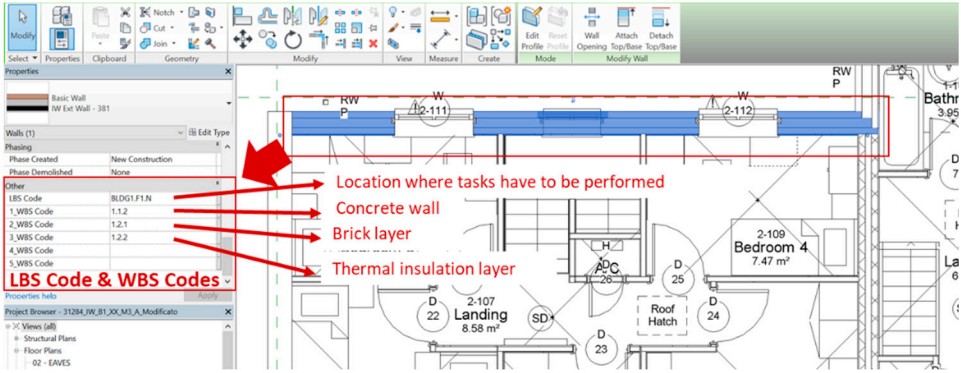

**Figure 5.** LBS and WBS codes displayed in the proprieties of a wall object.

*4.2. Augmented Reality for Context-Aware Information in Specific Locations on Site*

Augmented reality (AR) technology enhances real environments with digital contents through the use of head-mounted mobile devices. AR allows users to interact with both the real and virtual objects by overlaying digital information and 3D objects on real objects. AR is a promising technology for improving visualizations of information directly on construction sites. With AR, it is possible to show as-planned and as-built projects and visualize the construction progress [42]. Meža et al. [43] found that AR could facilitate an understanding of project documentation and construction progress through 3D visualizations of models on site. Park et al. [44] and Kopsida and Brilakis [30] investigated the use of AR in terms of monitoring project progress and comparing it to as-planned schedule.

The AR4C application is intended mainly for site managers and workers, and is envisioned as a tool to provide these individuals with context-specific information anytime and anywhere on a construction site. The AR4C application is a prototype version and was developed for the Android smartphone Lenovo Phab 2 Pro, since it integrates Google Project Tango technology like motion tracking and depth perception. Motion tracking compares images from big fish-eye cameras against movement data from the inertial measurement unit. The camera is used to track features of the real world like edges and corners as they move from frame to frame. Google Tango can perform measurements 100 times a second. A time-of-flight (ToF) camera and an IR projector are used for depth perception. Tiny points of the IR projector are sent out and reflected into the ToF camera, which calculates the distance via the travel time. This technology gives the ability to detect the device's position relative to the world around it with high precision and accuracy. These two features, accuracy and precision, are mandatory for displaying 3D models in AR, which are superimposed on the real world. Through the use of AR, users are able to interact with objects of the 3D model and the embedded information.

To develop a prototype of the AR4C application, Unity [45] was chosen as a main development environment, which is a real-time 3D platform for visualizing interactive 3D models and virtual experiences. To enable the visualization of the 3D model and information relative to construction tasks, it was necessary to interconnect different software and exchange multiple data. From Autodesk Revit, two files are exported: (a) a 3D model via an .fbx file and (b) its metadata via an .xml file, using the ACCEPT XML plugin. The .fbx file is first imported into Autodesk 3ds Max [46] to manage the model organization by entities, in order to maintain the same IDs of 3D objects in Unity as in Autodesk Revit. Afterwards, the geometry is exported as an .fbx file and imported directly to Unity. In Unity, the geometry of the 3D model is set-up, including position, scale, materials and physical characteristics (rigidbody, colliders, prevention of walking through walls, etc.). The .xml file with metadata is imported directly into Unity as well. It contains information such as geometrical data of objects (length, width, height of building components, quantity), object materials and IDs of objects and their WBS and LBS codes. The structure of an .xml file is pre-defined. Not all data included in this structure are needed for the correct functionality of the AR4C application. The required data are parsed from the .xml file by means of layers and XML Parser scripts, which extract data from

the 'Elements' and 'Types' of the .xml file structure and transform them into structures that are more suitable for data management in the application. From 'Elements', it is possible to get IDs of 3D objects in Autodesk Revit, and geometrical data such as 'width, 'height and 'type. Moreover, IDs of building components in Unity correspond to IDs of 'Elements' (e.g., <Id>336001</Id>) in the .xml file. 'Types' includes information related to the product and its technical data, as well as WBS and LBS codes. To link these data to building components in Unity, it is important to use the 'TypeID' code (e.g., <TypeId>337447</TypeId>). Figure 6 shows an example of the data structure included in the 'Elements' and 'Types' of the .xml file.

```
ELEMENTS>
<ELEMENT>
<Id>336001</Id>
<TypeId>337447</TypeId>
<Parameters>
<Parameter Value="-2000014" Id="-1140363" Name="Category"/>
<Parameter Value="1.1300" Id="-1010301" Name="Width"/>
<Parameter Value="1.9500" Id="-1010300" Name="Height"/>
<Parameter Value="337447" Id="-1002050" Name="Type"/>
<TYPES>
<TYPE>
<Id>337447</Id>
<Name>Finestra Wolf_ holz/alu 85c</Name>
<MaterialsIds>[26;349412]</MaterialsIds>
<Parameters>
  <Parameter Value="holz/alu 85c" Id="-1010109" Name="Model"/>
  <Parameter Value="Wolf Fenster" Id="-1010108" Name="Manufacturer"/>
  <Parameter Value="https://www.wolf-fenster.it/produkte/fenster-design/classic-p14-23.html" Id="-1010104" Name="URL"/>
  <Parameter Value="Double glazing - 1/4 in thick - bluegreen/low-E (e = 0.05) glass" Id="-1005437" Name="Analytic Construction"/>
  <Parameter Value="0.4500" Id="-1005433" Name="Visual Light Transmittance"/>
  <Parameter Value="0.2700" Id="-1005432" Name="Solar Heat Gain Coefficient"/>
  <Parameter Value="0.5032" Id="-1005431" Name="Thermal Resistance (R)"/>
  <Parameter Value="1.9873" Id="-1005430" Name="Heat Transfer Coefficient (U)"/>
  <Parameter Value="Finestra Wolf_ holz/alu 85c" Id="-1002001" Name="Type Name"/>
  <Parameter Value="0.0500" Id="337436" Name="Thickness"/>
  <Parameter Value="1.2.1" Id="349407" Name="Codice WBS"/>
  <Parameter Value="1.0000" Id="349629" Name="Uf"/>
  <Parameter Value="1.0000" Id="349631" Name="Ug"/>
```

**Figure 6.** Data structure of the 'Elements' and 'Types' in the .xml file.

To manage data imported into the AR4C application, it was necessary to create several components in Unity (Figure 7). So far, the following components have been created: (a) 3D model management, (b) data management, (c) Graphical User Interface (GUI) management and (d) a location plugin. The '3D model management' component manages the import of the .fbx file in order to make further use of the 3D model. Via scripts, relevant information is extracted from the model and is stored in the application in apposite data structures to categorize each building component and have access to their information. The 'data management' component manages data from both the .fbx file and .xml file generated by the ACCEPT XML plugin in Revit. All this information managed by components '3D model management' and 'data management' is stored in tailor-made data structures of the application, which are used by different services of the AR4C application. The component 'GUI management' allows the visualization of 3D models and related information in AR within the graphical user interface (GUI). The 'location plugin' component manages the location system, using sensory information collected by the mobile device to give location awareness codes to the application. The Lenovo Phab 2 Pro device is able to calculate its spatial position with great precision; however, this functionality cannot be directly correlated to locations defined by LBS codes. For this reason, the use of Bluetooth low energy (BLE) beacons (Estimote beacons) has been investigated in order to offer contextualized information on tasks and instructions. By applying beacons and naming (ID name) them with LBS codes, the AR4C will be able to recognize which beacon is approaching and, as a result, will display

the scheduled task as well as visualize information on task progress and performance. This approach is discussed in part in [47].

The components that have to be developed in order to provide all planned functionalities are scheduling management and dashboard management. The first component aims to display the construction progress and performance KPIs in AR for tasks in specific locations, and to highlight building components of the 3D model with different colors that correspond to task progress status. This functionality can be described by the following use case: When a user is approaching a construction location (e.g., F1.U2), the AR4C application retrieves LBS code information from the BLE beacon. This will trigger the searching process for all running tasks scheduled in this location. The name of tasks will be displayed (e.g., wall installation) with their current performance and progress KPIs. To enable the searching process for KPIs of construction tasks in real-time, a database is needed to store the information. There are plans to integrate the Firebase SDK database, where updated versions of the .xml file with construction progress and performance KPIs are systematically uploaded. Once WBS/LBS codes of the running task are found by the application, the application searches the IDs of building components in the 3D model with the same WBS/LBS code and displays progress statuses for them.

The 'dashboard management' component should interact with Microsoft Power BI [48] to provide an interactive dashboard and display construction progress and performance KPIs for selected construction tasks in specific locations. The integration has yet to be investigated, but it could be done via a URL link to Power BI metrics published on a web page. Table 1 shows KPIs that will be monitored and visualized to help the site manager in identifying problems early so that they are able to make conscious decisions on time and prevent scheduling deviations.

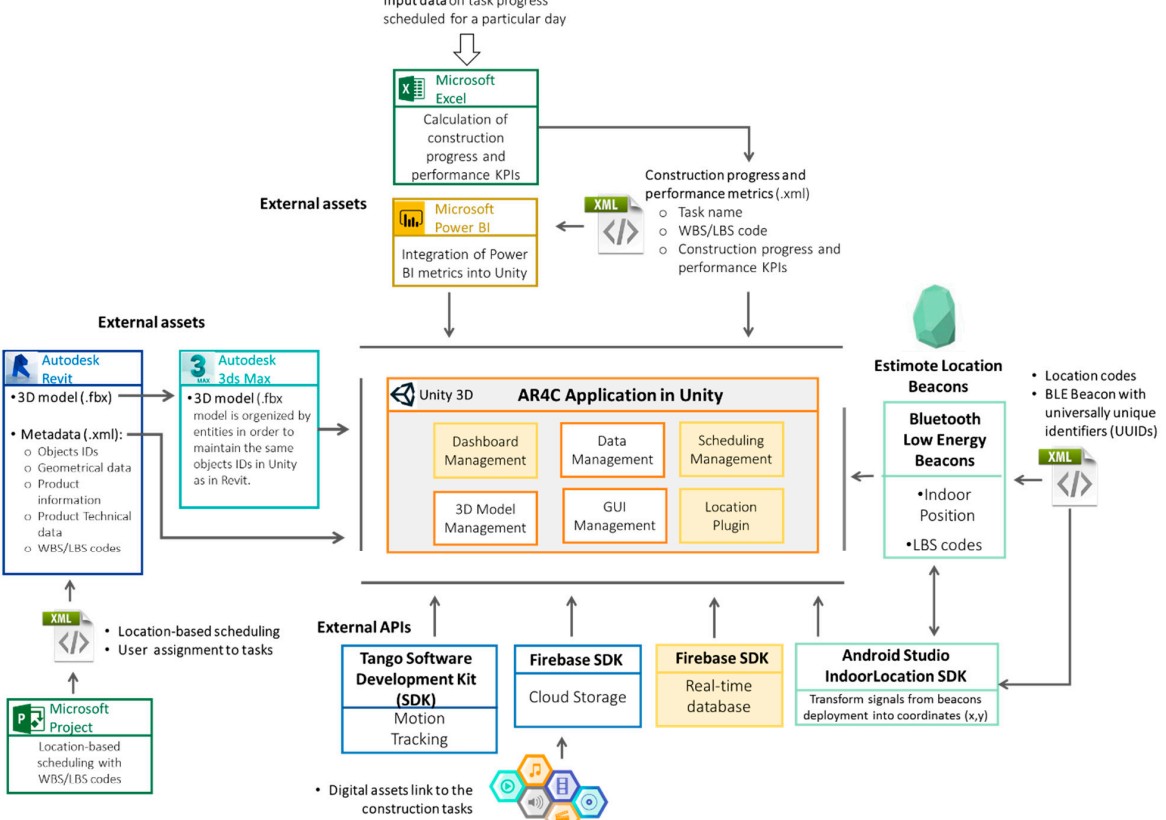

**Figure 7.** System architecture of the AR4C application.

**Table 1.** List of Key Performance Indicators that will be monitored and visualized through the AR4C application.

| Construction Progress and Performance KPIs | Definition | Implemented (I) or Planned (P) KPIs |
|---|---|---|
| **current progress** (CP) | **CP** [%] is the relation of the pitch content of a single activity to the overall pitch content of the whole workflow. | I |
| **Performance ability ratio** (PAR) | **PAR value** [-] is the ratio of the defined content of 1 pitch to the actual measured progress on site. Value > 1 indicates a lack of performance with respect to the expected performance. Value = 1 means that the foreseen goal has been met. Value < 1 refers to a more powerful performance than expected. Ranking activities regarding this criteria provides perception towards the improvement potentials of a single activity. | I |
| **Reason for non-completion** (RNC) | **RNC** [-] states a root cause for activities not completed on time. It allows the analysis of poorly running task. | I |
| **Percent plan completed** (PPC) | **PPC** [%] is the ratio of fulfilled assignments (achieved goals) to the total number of assignments scheduled for a particular day. If the goal is achieved PPC value is 100%; if not, it is 0%. The PPC value provides information regarding the reliability of the scheduling and the smoothness of the workflow. | I |
| **delay indicator** (DI) | **DI** [days] is the difference between planned working days and remaining days. | I |
| **extra effort** (EE) | **EE** [days] is the sum of the delay indicator for each activity in a task or tasks in a work package. | I |
| **quality gate** (QG) | **QG** [-] is the number of fulfilled quality checklists out of the total number of checks assigned to a task. | P |
| **construction errors** (CE) | **CE** [-] is the number of construction errors detected during inspections by the site manager. | P |
| **extra costs** (EC) | **EC** [€] is an additional cost calculated as a multiplication of extra effort required, expressed in days per man-hour cost rate. | P |

KPIs are calculated in Excel [49] by providing 'input data'—the percentage of completed activities scheduled on a specific day for a particular task in a specific location. To be able to provide reliable data on the construction progress and performance, construction works are monitored on a daily basis. Data are imported into Power BI via an Excel file, which is visualized on a dashboard through charts and widgets. It reports relevant information at a glance, including KPIs to meet the predefined objectives of the project, which helps to keep the project within a schedule and budget, and to reduce the intensive labor effort that is typically required in the manual reporting process.

Finally, external application programming interfaces (APIs) have to be integrated into Unity, including: (a) Tango Software Development Kit (SDK), which provides different features used to gather information on the device (smartphone) position and orientation, as well as to interact with it; (b) Firebase SDK, which stores digital assets such as images, videos, checklists, drawings and messages, and which allows the AR4C application to access these assets as needed while acting as a digital data repository of the application; (c) Estimote Indoor Location SDK, which allows real-time beacon-based mapping of indoor location.

## 5. Results

The AR4C application is under development and currently has reached maturity at level 4, according to the technology readiness level (TRL) scale. This means that basic technological components were developed, integrated and tested in a laboratory environment. Table 2 shows the implemented functionalities as well as planned functionalities for further development.

**Table 2.** List of main implemented and planned functionalities of the AR4C.

| AR4C Functionalities | Description | Implemented (I) or Planned (P) Functionalities |
|---|---|---|
| Navigate 3D Model | The user navigates the 3D model in the application by walking in the real environment. The model remains aligned with the surroundings, since the application uses motion tracking and depth perception technology (Figure 8a). | I |
| Filter 3D Model | The user can enable and disable different layers (groups of elements), and therefore sees only objects of interest (Figure 8a). | I |
| Select an element and visualize its information | The user can touch every element of the 3D model and read information from them. The selected element is colored in green (Figure 8b). | I |
| Read geometry information | The user can visualize geometrical and technical data of a selected component. Information is retrieved from the .xml file generated in Autodesk Revit (Figure 8b). | I |
| Consult task list | The user can consult a list of tasks currently available in a specific location. By clicking on the task, the information panel appears. It provides the following types of information (Figure 8c): (a) a step-by-step tab that shows the steps that should be followed by a worker in order to perform a task; (b) an instructions tab, which shows a document with installation procedures that can be scrolled down; (c) a construction details tab, which contains construction drawings and details; (d) a checklist tab, which contains a quality checklist that should be filled out by a worker at the end of the task. | partially I |
| Upload/read note | The user can type/read a note related to a selected component and upload/download it to/from the shared database by touching a button. | I |
| Display KPIs (planned functionality) | The user can display construction performance and progress KPIs for a task in a specific location by visualizing the Power BI dashboard. | P |
| Visualize task progress status by highlighting elements of the 3D model (planned functionality) | When the users select a task status in a specific location, all building elements of the 3D model are colored according to the status (red = behind schedule; green = on schedule; blue = ahead of schedule) and KPIs from Table 1 are reported as well (Figure 8d). | P |
| Recognize LBS codes and provide task list (planned functionality) | The AR4C application retrieves information using an LBS code from BLE beacons when the user is approaching a location on site. This will trigger the searching process for all running tasks scheduled in this location. | P |

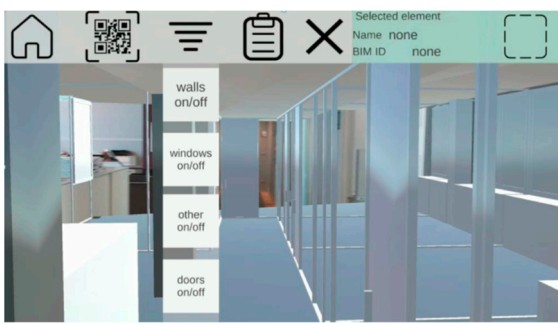

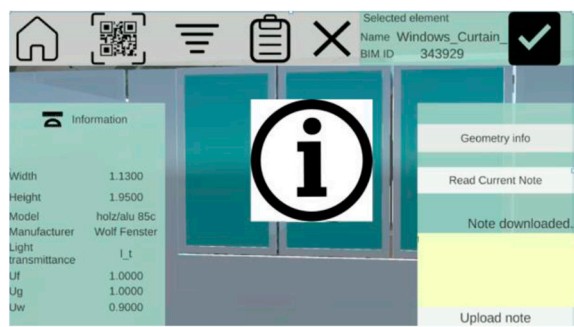

(a)                                                    (b)

**Figure 8.** *Cont.*

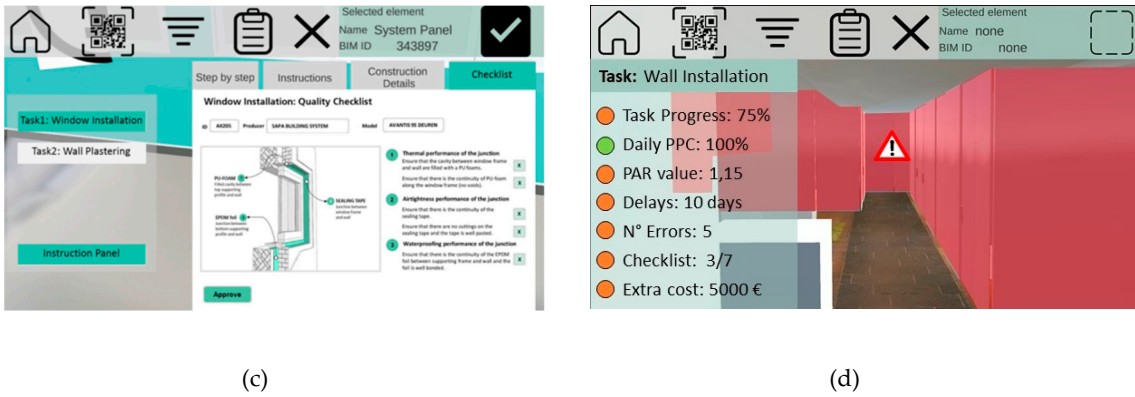

　　　　　　　　　　　　　　(c)　　　　　　　　　　　　　　　　　　　　　　　　　　(d)

**Figure 8.** Functionalities of the AR4C: (**a**) 3D model navigation and object filtering; (**b**) geometrical and technical data of selected objects; (**c**) task list in a specific location with related information on scheduled work; (**d**) progress of construction task displayed on a 3D model with related KPIs.

## 6. Discussion

The AR4C application is a prototype solution. Different components and functionalities have already been implemented into the application; however, components related to construction project controls are stand-alone components, and have yet to be integrated. The Excel application for construction planning and controlling, as well as the dashboard component, were tested by end users (Epitessera Architects) during the construction of the English School in Nicosia (Cyprus). Tests were carried out for concrete slab casting, and the testing phase achieved promising results. According to the architect and site manager, the LBMS combined with 3D visualization, as well as construction progress and performance metrics, reduced the time needed by more than 50% compared with conventional procedures. In addition, the automated extraction of intuitively comprehensible project KPIs and their BIM-related visualization in the dashboard have been appreciated as a very powerful tool.

The AR4C was tested with its implemented functionalities in a laboratory environment and in two real buildings (one of them during construction) to validate the alignment of the 3D model superimposed onto the real building. The testing utilized a definition of the starting point (x, y, z) for the application in the real world. The same position was applied to a Tango camera in Unity. Such an approach should allow the perfect alignment of both real and virtual worlds. However, the results of this testing showed several alignment errors of the 3D building model in AR. The model was not always perfectly superimposed onto the real building. The difference varied between approx. 0.4–1 m. After testing, it was assumed that the lack of perfect alignment was caused by: (a) incorrect setting of the field of view (FOV) of the device's camera and the FOV of the Tango camera; and/or (b) problems with the positioning of the device at the starting point. The initial error related to the starting point propagates further computation errors of the device position. Another alignment error occurred when the user reached the surfaces of virtual objects. In this situation, the model visualization was blocked, and the computation of the device position was affected, resulting in model misalignment. This testing indicated ways to improve the accuracy of the AR experience.

Moreover, functionalities and user interface were tested in focus groups with users. The groups provided feedback and recommendations for the further developments. Almost all respondents considered it very likely that the information provided by the AR4C prototype would allow a faster access to relevant information on site, and could improve the productivity of the construction process if the monitoring of construction works were be implemented. For this reason, the integration of construction progress and performance component results is very important. The concept of the dashboard and progress visualizations on the 3D model in AR have been proposed as future implementations. This should save time for site managers in reviewing reports and calculating data, while providing a better understanding of the project and allowing site managers to make fast decisions and corrective actions.

Finally, the validation of the AR4C on construction sites is needed. There are plans to test the visualization of the 3D model in AR again, using different phases of construction to evaluate which context situations create major problems. Also, the proposed monitoring method according to LBMS will be tested for certain construction tasks in order to evaluate how this method can improve the productivity of the construction process and define the benefits and obstacles faced in adopting these methods. In adopting new technologies and methods, the human factor is fundamental. Therefore, user acceptance and the user experience will be considered by involving people from construction sites and asking them to use AR4C applications in different scenarios that simulate real situations on site.

## 7. Conclusions

This research paper described methodologies and enabled technologies, as well as implemented and planned functionalities of a BIM-based AR application combined with a location-based management system, referred to as the AR4C. The AR4C is a mobile field application for site managers to automate their daily work and improve construction performance. The application provides users with context-specific information related to construction projects using ARlike 3D models, geometrical and technical features of building components and materials, lists of construction tasks, installation procedures, construction checklists and construction progress and performance metrics. In particular, the methodology to integrate a location-based management system (LBMS) into BIM software (Autodesk Revit) and AR platform (Unity 3D) was discussed. Since it is not possible to define locations according to LMBS in Autodesk Revit, a mechanism to assign LBS and WBS codes to building elements and materials was introduced. This was fundamental in order to be able to manage objects of 3D models and visualize task in specific locations, their instructions and progress/performance KPIs. To link scheduled tasks in a specific location to building objects, it was necessary to develop scripts that correlated object IDs of the 3D models with WBS and LBS codes from data files that can be imported into Unity.

The novelty of the AR4C application consists of creating a unique field application that is able to easily detect scheduling deviations by visualizing construction progress in AR, and to provide daily progress and performance data of construction work, as well as context-specific information/documents on scheduled tasks. The advantage of this technology is to monitor and control construction tasks in different locations while the site manager is performing field inspection. This means that the site manager does not have to analyze huge amounts of data in a back office to extract information on construction status and performance. Using the AR4C, this information is displayed automatically on site, a feature which differs from the other commercial solutions available on the market. The limitation of the proposed solution is that the AR4C application is a prototype, and the system has not yet been fully integrated. Functionalities related to the visualization of construction progress and performance KPIs, as well as the dashboard view, have not been implemented yet. The system framework has been developed, but further research is needed to integrate different components and test the application on construction sites while considering its different phases. It will be important to define the technological limitation of the device when it is tracking features of early construction phases (e.g., excavations, foundation) with few reference points. Moreover, alignment errors of the virtual model with real environments have to be solved. Finally, the LMBS requires the adoption of specific rules for 3D modeling using BIM software. 3D models should be created ad hoc for construction sites in a way that considers construction locations and inserts WBS and LBS codes properly. Therefore, BIM modelers should follow specific modeling guidelines. To provide site managers with construction progress and performance KPIs, it is necessary to dispose of baseline productivity data for all planned construction works. This requirement could be a serious obstacle for many construction companies that schedule works based on their own experiences.

**Author Contributions:** Conceptualization, J.R.; Methodology, J.R.; Validation, J.R.; Writing—original draft preparation, J.R., D.T.M. and M.R.; Writing—review and editing, D.T.M. and M.R.; Visualization, J.R.; Supervision, D.T.M. and M.R.

**Funding:** This research received no external funding. This research was an internal research of Fraunhofer Italia Research, which is currently being developed within the PhD program Sustainable Energy and Technologies at the Free University of Bozen-Bolzano. The basis of the AR4C research project was initiated by the ACCEPT project and funded by the European Commission within the Horizon 2020 Framework Programme.

**Acknowledgments:** The authors (Grant Agreement No. 636895) would like to thank Andrei Popescu for IT development of the AR4C. The development part was described in his bachelor's thesis at the Free University of Bozen-Bolzano: Context-Aware Information Delivery for the Construction Site Using a Mixed Reality Mobile Application. The authors gratefully acknowledge the contributions of the other researchers involved in the AR4C project, especially Christoph Paul Schimanski and Carmen Marcher from Fraunhofer Italia Research for the collaboration in implementing LBMS in BIM software and the creation of the Excel program for calculating construction progress and performance KPIs; and Alice Schweigkofler from Fraunhofer Italia Research for conducting the research on integration of BLE beacons in AR4C. The authors would like to thank the European Commission for their funding of the ACCEPT project within the Horizon 2020 Framework Program (Grant Agreement No. 636895). The authors also acknowledge the contributions of the partners from the ACCEPT project that created the basis for the AR4C project research, especially Peter Leo Merz from TIE Germany, who was responsible for the implementation of lean construction methods in Profile Nexus; Vincent Delfosse, Hatem Bejar, Anabelle Rahhal, Pierre Leclercq from LUCID Lab for the development of the Dashboard; Pablo Gilabert from Cype Soft for the development of the XML plugin for Autodesk Revit; and George Georgiou and Elena Parouti from Epitessera Architects for the testing of the dashboard and construction controlling tool.

**Conflicts of Interest:** The authors declare no conflict of interest. The funders had no role in the design of the study; in the collection, analyses, or interpretation of data; in the writing of the manuscript, or in the decision to publish the results.

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
