# Peer review of "BIM-based and AR Application Combined with Location-Based Management System for the Improvement of the Construction Performance"

_buildings, doi:10.3390/buildings9050118_

Reviewer 1 Report

The subject of the paper is really original and can be quite successful among the readers of the Journal.

The comparative and experimental methodology is widely shared, through the whole process of research development.

In the Discussion, the authors refer to some errors that need additional insights and experimentation. These errors could be described and illustrated in more detail. 

In this way, the Reader will be able to appreciate the entire experimental course, and the authors still lay the basis for a subsequent paper on the research improvements. 

Author Response

Dear Reviewer, 

I would like to thank you for your time and comments. I tried to include all your suggestions in the attached manuscript.  As you recommended, I provided additional insight related to the testing phase. I hope it will meet  your expectation. 

From line 376

Afterwards, the system integration of the AR4C application was described as well as its functionalities. Implemented functionalities were tested in a laboratory environment and in two real buildings (one of them during the construction) to validate the alignment of the 3D model superimposed onto the real building. The testing consisted in the definition of the starting point (x, y, z) for the application in the real world. The same position was applied to Tango camera in Unity. Such approach should allow the perfect alignment of both real and virtual worlds. However, the results of this testing showed several alignment errors of the 3D building model in AR. It was not always perfectly superimposed onto the real building. The difference was variable and approximately varied between 0,4 and 1 meters. After testing the following assumption were stated. The lack of perfect alignment is caused by: a) incorrect setting of the Field of View (FOV) of the device's camera and the FOV of the Tango camera; b) problems with the perfect positioning of the device at the starting point. The initial error related to the starting point propagates further computation errors of the device position. Another alignment error was occurring, when the user was reaching the surface of virtual objects. In this situation, the model visualization was blocked and the computation of the device position was affected, resulting in model misalignment. This testing provided indications to improve the accuracy of the AR experience.

Best regards, 

 Julia Ratajczak

Reviewer 2 Report

Authors proposed the application for the monitoring and controlling of project performance and progress according to lean construction methods as well as visualization of construction progress and performance in a specific location using augmented reality. The developments presented are useful for site managers as proposed mobile field application let to automate their daily work and improve construction performance. The proposed paper fit the journal topics. Some comments arose during the review.

The reviewer invites you to find a way to take into account these comments:

The introduction does not explain the structure of the paper. Provide a detailed explanation of the structure.

The captions of Table 1 and Table 2 indicate the same – list of main functionalities, however, present different information. Table 1 presents the KPIs, while Table 2 presents the list of functionalities. Revise the captions.

Table 1 presents the “developed and planned” KPIs, and respectfully, Table 2 presents the “implemented and planned” functionalities. These “developed and planned” and “implemented and planned” positions have to be clearly indicated in tables. I suggest to include an additional column in both tables and indicate if exact KPI element is “developed” or “planned” (Table 1), and if exact functionality “implemented” or “planned” (Table 2).

The section Conclusions should be included. There is a need to clarify the novelty of the material presented, knowledge gaps filled by the research findings. In Conclusions, provide clearly the advantages and disadvantages as well the limitations of the proposed approach, including a brief conclusion on developed and implemented functionalities of the product.

Authors mentioned that research activities on the AR4C project initiated in the European project - ACCEPT 156 (www.accept-project.com). In such a case, the acknowledgments to the funding body have to be provided at the end of the paper.

In addition, I want to bring to your attention the very significant nonconformity that indisputably must be corrected. The manuscript does not include a sufficient number of references to scientific peer-reviewed journals. Only 27 % of references are journal papers, and the latter are oldish. Rearrange the manuscript’s reference list by adding references from English peer-reviewed journals. Use the most recent scientific articles.

Author Response

Dear Reviewer, as you recommended, I considered your comments and suggestions. I hope these modifications will meet  your expectations.

Best regards,

Julia Ratajczak

Point 1:

The introduction does not explain the structure of the paper. Provide a detailed explanation of the structure.

Response 1:

Based on the literature review, it has emerged that an application, which is able to merge BIM, AR and Lean functionalities is required to maximize the improvement of construction processes on site. The authors decided to investigate the potential of an AR field application, that can provides users with the visualization of BIM model, project information and information on construction progress and performance according to Lean Construction. In this paper, the authors present a solution of BIM-based and AR application for site managers and workers that is covering aforementioned gaps. The authors focus mainly on the description of methodologies and technologies used during the development of the application and its main functionalities as well as system integration. Firstly, the technological solution of the field application is explained (Section 2). To provide all functionalities, which are listed in Figure 1, several technologies and methodologies have been integrated into the application. Their description and implementation methods are discussed in section 3. Moreover, functionalities of the application are described in this sections as well. In section 4, the preliminary tests of the application in a laboratory environment and in real buildings were discussed. Finally, as a conclusion, the authors outline the novelty of the application and gaps that covers in relation to the other solutions. Advantages and limitation of the proposed solution are highlighted as well.

Point 2:

The captions of Table 1 and Table 2 indicate the same – list of main functionalities, however, present different information. Table 1 presents the KPIs, while Table 2 presents the list of functionalities. Revise the captions.

Table 1 presents the “developed and planned” KPIs, and respectfully, Table 2 presents the “implemented and planned” functionalities. These “developed and planned” and “implemented and planned” positions have to be clearly indicated in tables. I suggest to include an additional column in both tables and indicate if exact KPI element is “developed” or “planned” (Table 1), and if exact functionality “implemented” or “planned” (Table 2).

Response 2:

The caption of Table 1 has been corrected.

The correct version is:

Table 1: List of KPIs that will be monitored and visualized through the AR4C application.

I have added the third column as you suggested to indicated planned/implemented functionalities and KPIs. Now it is more clear. Thank you!

Point 3:

The section Conclusions should be included. There is a need to clarify the novelty of the material presented, knowledge gaps filled by the research findings. In Conclusions, provide clearly the advantages and disadvantages as well the limitations of the proposed approach, including a brief conclusion on developed and implemented functionalities of the product.

Response 3:

The section Conclusion has been added to the manuscript.

This research paper describes methodologies, enabled technologies as well as implemented and planned functionalities of the BIM-based and AR application combined with Location-based Management System, so-called AR4C. It is mobile field application for site managers to automate their daily work and improve the construction performance. This application provides users with context-aware information in AR related to the construction project like 3D model, geometrical and technical features of building components and materials, list of construction tasks, installation procedures and construction checklists as well as construction progress and performance metrics. In particular, in this paper the methodology to integrate Location-based Management System (LBMS) into BIM software (Autodesk® Revit™) and AR platform (Unity 3D) was discussed. Since Autodesk® Revit™ is not possible to define locations according to LMBS, a mechanism to assign LBS and WBS codes to building elements and materials was introduced. It was fundamental in order to be able to manage objects of 3D models and visualize task in specific locations, their instructions and progress/performance KPIs. To link scheduled tasks in a specific location to building objects, it was necessary to develop scripts that correlates objects IDs of the 3D model with WBS and LBS codes from data files that are imported to Unity.

The novelty of the AR4C application consists in creating a unique field application that is able to detect easily scheduling deviation by visualizing construction progress in AR, to provide daily progress and performance data of construction work as well as to provide context-aware information/documents on scheduled tasks. The advantage of this solution is to control construction tasks in different locations, while site manager is performing his field inspection. It means that the site manager does not have to analyze huge amount of data in the back office to extract information on the construction status and performance. Using the AR4C, this information is displayed automatically on site. This feature is differing AR4C application from the other commercial solutions available on the market. The limitation of the proposed solution is that up to now the AR4C application is a prototype and the system is not fully integrated. Functionalities related to the visualization of construction progress and performance KPIs as well as the dashboard view have not implemented yet. The system framework has been developed, but the further research is needed to integrate different components and test the application on the construction site, considering its different phases. It will be important to define technological limitation of the device, when it is tracking features of the early construction phase (e.g. excavations, foundation) with few reference points. Moreover, alignment errors of the virtual model on the real environment has to be improved. Finally, the LMBS implies to adopt specific rules for the 3D modelling in BIM software. 3D models should be done ad hoc for the construction site in a way to consider construction locations and to insert properly WBS and LBS codes. Therefore, BIM modelers should follow specific modelling guidelines. To provide site manager with construction progress and performance KPIs is necessary to dispose of baseline productivity data for all planned construction works. This requirement could be a serious obstacle for many construction companies that schedule works based on their experience.

Point 4:

Authors mentioned that research activities on the AR4C project initiated in the European project - ACCEPT 156 (www.accept-project.com). In such a case, the acknowledgments to the funding body have to be provided at the end of the paper.

Response 4:

Thank you for noticing this missing acknowledgment.

I have added it:

The authors would like to thank the European Commission for their funding of the ACCEPT project within the Horizon 2020 Framework Program (Grant Agreement No. 636895).

Point 5:

In addition, I want to bring to your attention the very significant nonconformity that indisputably must be corrected. The manuscript does not include a sufficient number of references to scientific peer-reviewed journals. Only 27 % of references are journal papers, and the latter are oldish. Rearrange the manuscript’s reference list by adding references from English peer-reviewed journals. Use the most recent scientific articles.

Response 5:

I have actualized reference with peered-reviewed journals. The literature review is done with articles in average between 2013-2018. There are still few older articles, because they are referring to specific research projects. Now, 47% of references are journal papers.

Beyond journal papers there are some references to books and PhD thesis, statistic reports/web page reports, conference proceedings and software. I hope it will meet your requirements. If you consider that the amount of journal references is still low, I am willing to make further adjustments.

The current Reference list is as follows:

1.        McKinsey Global Institute. Reinventing construction: A route to higher productivity. McKinsey & Company, February 2017; pp. 2–10.

2.        Andri´c, J.M.; Mahamadu, A.M.; Wang, J.; Zou, P.X.W. The cost performance and causes of overruns in infrastructure development projects in Asia. Journal of Civil Engineering and Management 2019, 25, pp. 203–214.

3.        McKinsey Global Institute. The construction productivity imperative. McKinsey & Company, June 2015; pp. 3–9.

4.        Katre, V.Y., Ghaitidak, D.M. Elements of Cost and Schedule Overrun in Construction Projects. International Journal of Engineering Research and Development 06/2016, 12( 7), pp.64–68.

5.        Hussin, J.M.; Rahman, I.A.; Memon, A.H. The way forward in sustainable construction: Issues and challenges. International Journal of Advances in Applied Sciences 2013, 2, pp. 15–24.

6.        Aziz, R.F., Hafez, S.M. Applying lean thinking in construction and performance improvement. Alexandria Engineering Journal 2013, 52, pp. 679–695.

7.        KPMG International. Global construction survey 2015: Climbing the curve. PMG International Cooperative, March 2015; pp. 2–3.

8.        Memon, A. H.; Rahman, I. A.; Aziz, A.A.A. The cause factors of large project’s cost overrun: a survey in the southern part of peninsular Malaysia. International Journal of Real Estate Studies 2012, 7(2)

9.        Salehi, S.A, Yitmen, I. Modeling and analysis of the impact of BIM-based field data capturing technologies on automated construction progress monitoring. International Journal of Civil Engineering, 12/2018, 16(12), pp 1669–1685.

10.      Maalek, R.; Sadeghpour, F.  Accuracy assessment of Ultra-Wide Band technology in tracking static resources in indoor construction scenarios, Automation in Construction 3/2013, 30, pp.170–183.

11.      Dallasega, P., Rauch, E., Frosolini, M. A Lean Approach for Real-Time Planning and Monitoring in Engineer-to-Order Construction Projects. Buildings 2018, 8(3), 38..

12.      Lin, J.J., Golparvar-Fard, M., Visual Data and Predictive Analytics for Proactive Project Controls on Construction Sites. In: Smith I., Domer B. (eds) Advanced Computing Strategies for Engineering. EG-ICE 2018. Lecture Notes in Computer Science, vol 10863, Springer, Cham.

13.      Love, P.E.D., Smith, J., Ackermann, F., Irani, Z., Teo, P. The costs of rework: insights from construction and opportunities for learning. Production Planning & Control 2018, 29(13), pp.1082–1095.

14.      Zavadskas, E.K.; Vilutienė, T.; Turskis, Z.; Šaparauskas, J. Multi-criteria analysis of Projects’ performance in construction. Archives of Civil and Mechanical Engineering 2014, 14(1); pp. 114–121.

15.      Yi, W.; Chan, A.P.C. Critical Review of Labor Productivity Research in Construction Journals. Journal of Management in Engineering 2014, April; pp. 214–225.

16.      Dainty, A.; Moore, D.; Murray, M. Communication in Construction: Theory and practice, Taylor & Francis, New York, NY, USA, 2006; pp. 19–52.

17.      Love, P.E.D., Lopez, R., Kim, J.T., Kim, M.J. Influence of Organizational and Project Practices on Design Error Costs. Journal of Performance of Constructed Facilities 04/2014, 28(2).

18.      Kenley, R.; Seppänen, O.  Location-based Management System for Construction: Planning, Scheduling and Control. Spon Press, London, UK and New York, USA, 2010.

19.      McKinsey Global Institute. Digital Europe: Pushing the frontier, capturing the benefits. McKinsey & Company June 2016; pp. 7–22.

20.      BIM360 (2019) Autodesk (https://www.autodesk.com/bim-360/).

21.      Connected BIM (2019) Oracle Aconex (https://help.aconex.com/aconex/our-main-application/aconex-release-notes-updates/introducing-connected-bim-expanded-aconex-mobile-suite-improved-process-management).

22.      Latista (2015) Oracle (https://www.microsoft.com/en-us/p/oracle-latista-field-management/9nblggh2spn8?activetab=pivot:overviewtab).

23.      TwinBIM (2017) Dalux (https://www.dalux.com/dalux-field/twinbim/).

24.      Vico Office (2012) Trimble (https://connect.trimble.com/feature/vico-office.html).

25.      VisiLean Software (2019) VisiLean (http://visilean.com/).

26.      Sacks, R.; Barak, R; Belaciano, B.; Gurevich, U.; Pikas, E. KanBIM Workflow Management System: Prototype implementation and field testing. Lean Construction Journal 2013; pp. 19–35.

27.      Mccoy, A.P., Golparvar-Fard, M., Rigby , R.T. Reducing Barriers to Remote Project Planning: Comparison of Low-Tech Site Capture Approaches and Image-Based 3D Reconstruction. Computer-Aided Civil and Infrastructure Engineering 01/2012, 20(1), pp.

28.      Kopsida, M.; I. Brilakis. Markerless BIM Registration Methods for Mobile Augmented Reality-Based Inspection. Proceedings of the 11th European Conference on Product and Process Modelling (ECPPM 2016), Limassol, Cyprus, 7-9 September; pp. 1631–1636.

29.      Ratajczak, J.; Marcher, C.; Riedl, M.; Matt, D.T.; Mayer, N.; Sánchez, J.; Georgiou, G.; Rahhal, A.; Page, J.; Perez Alonso, J.M.; Chepegin, V.; Brancart M. Digital Tools for the Construction Site. A Case Study: ACCEPT Project. Proceedings of the Joint Conference on Computing in Construction (JC3) LC32017: Volume I, Heraklion, Greece, 4-12 July 2017; pp. 981–988.

30.      National Building Information Model Standard Project Committee. Available online: www.nationalbimstandard.org/faqs (accessed on 15/03/2019).

31.      Eastman, C. M.; Teicholz, P.; Sacks, R.; Liston, K. BIM handbook: A guide to building information modeling for owners, managers, architects, engineers, contractors, and fabricators, Wiley, Hoboken, N.J., 2008, p. 1.

32.      Sacks, R.; Koskela, L.; Dave, B.A.; Owen, R. Interaction of Lean and Building Information Modeling in Construction. Journal of Construction Engineering and Management 2010, 136(9); pp. 968–980.

33.      Dave, B. . Developing a construction management system based on lean construction and building information modelling. PhD Thesis, University of Salford, UK, 2013.

34.      Khanzode, A., Reed, D.; Fischer, M. Benefits and lessons learned of implementing Building Virtual Design and Construction (VDC) technologies for coordination of Mechanical, Electrical, and Plumbing (MEP) systems on a large Healthcare project. Electronic Journal of Information Technology in Construction 2008, 13, pp. 324-342.

35.      Khemlani, L. Sutter Medical Center Castro Valley: Case Study of an IPD Project, 2009. Available online: www.aecbytes.com/buildingthefuture/2009/Sutter_IPDCaseStudy.html (accessed on 15/03/2019).

36.      Deshpande, A., Salem, O.M., Filson, L.E. Miller, R.A. Lean Techniques in the Management of the Design of an Industrial Project. Journal of Management in Engineering 04/2012, 28(2), pp. 221–223.

37.      Seppänen, O. Empirical research on the success of production control in building construction projects. Doctoral thesis, Department of Structural Engineering and Building Technology, Faculty of Engineering and Architecture, Helsinki University of Technology, Espoo, Finland, 2009.

38.      Seppänen, O., Evinger, J. & Mouflard, C. Effects of the location-based management system on production rates and productivity. Construction Management and Economics 2014, 32(6), pp. 608-624.

39.      Ms Project (2017) Microsoft (https://products.office.com).

40.       Grubert, J., Langlotz, T., Zollmann, S., Regenbrecht, H. Towards Pervasive Augmented Reality: Context-Awareness in Augmented Reality. IEEE Transactions on Visualization and Computer Graphics 01/2017, 23(6), pp.1706-1724.

41.      Meža, S.; Turk, Ž.; Dolenc, M. Measuring the potential of augmented reality in civil engineering. Advances in Engineering Software 2015, 90; pp. 1–10.

42.      Park, C. S.; Lee, D. Y.; Kwon, O. S.; Wang, X. A framework for proactive construction defect management using BIM, augmented reality and ontology-based data collection template. Automation in Construction 2013, 33; pp. 61–71.

43.      Unity (2016) Unity (https://unity.com/).

44.      3ds Max (2017) Autodesk (https://www.autodesk.com/products/3ds-max/overview).

45.      Schweigkofler, A.; Pasetti Monizza, G.; Domi, E.; Popescu, A.; Ratajczak, J.; Marcher, C.; Riedl, M.; Matt, D. Development of a digital platform based on the integration of augmented reality and BIM for the management of information in construction processes. Proceedings of the IFIP 15th International Conference on Product Lifecycle Management. Special session 1.4 - Building Information Modeling, , Turin, Italy, 1-4 July, 2018; pp. 46–55.

46.      Power BI (2018) Microsoft (https://powerbi.microsoft.com).

47.      Excel (2017) Microsoft (https://products.office.com).

Round  2

Reviewer 2 Report

Accept in present form